# Musculoskeletal Disorder Burden and Its Attributable Risk Factors in China: Estimates and Predicts from 1990 to 2044

**DOI:** 10.3390/ijerph20010840

**Published:** 2023-01-02

**Authors:** Zeru Yu, Jingya Zhang, Yongbo Lu, Ning Zhang, Bincai Wei, Rongxin He, Ying Mao

**Affiliations:** 1School of Public Policy and Administration, Xi’an Jiaotong University, Xi’an 710049, China; 2School of Public Health and Emergency Management, Southern University of Science and Technology, Shenzhen 518055, China; 3Vanke School of Public Health, Tsinghua University, Haidian District, Beijing 100084, China

**Keywords:** musculoskeletal disorders, joinpoint regression, risk factors, age–period–cohort model, prediction, burden of diseases

## Abstract

Musculoskeletal disorders are one of the three major disabling diseases in the world. However, the current disease burden in China is not well-known. This study aimed to explore the burden and risk factors of musculoskeletal disorders in China from 1990 to 2019, predicting the incidence trend from 2020 to 2044. All data were extracted from the Global Burden of Disease Study 2019 (GBD 2019). Joinpoint regression and age–period–cohort (APC) models were selected to analyze the epidemic trend, and descriptive analyses of the time trends and age distributions of risk factors were performed. The Bayesian APC model was used to foresee the incidence trend from 2020 to 2044. The results indicated that the burden of musculoskeletal disorders is higher in women and older adults. Its attributable risk factors were found to be tobacco, a high body mass index, kidney dysfunction and occupational risks. In 2044, musculoskeletal disorders in China showed a downward trend for 35–59-year-olds and a slight upward trend for 30–34- and 65–84-year-olds. The 70–74 year age group saw the largest increase in incidence at 4.66%. Overall, the incidence increased with age. Therefore, prevention and control policies should focus on women and the elderly, and health interventions should be carried out based on risk factors.

## 1. Introduction

Musculoskeletal disorders, which have become increasingly severe worldwide in recent years [1], are defined as disorders of any type of discomfort that affect the motor organs, muscles, tendons, bones, cartilage and ligaments [2]. According to the International Classification of Diseases (ICD), musculoskeletal disorders refer to a condition related to musculoskeletal system or metabolic immune disease. ICD is an internationally unified disease classification method formulated by the WHO (World Health Organization). It classifies diseases into categories according to their etiology, pathology, clinical manifestations, and anatomical location, making them an orderly combination and expressed by coding methods. With age, musculoskeletal tissues exhibit increased skeletal fragility, the loss of cartilage elasticity, reduced ligament elasticity, the loss of muscle strength, and fat redistribution [3]. However, the importance of musculoskeletal disorders is often underestimated as they are rarely fatal [4]. This disease has become one of the most widespread problems for people in all countries due to the following reasons: low priority in disease prevention and control agenda, the wide range of risk factors, and the high cost of treatment, which seriously pose heightened threats to global public health [5]. Moreover, the incidence and death rates of this disease show vast geographical differences globally, especially between developing and developed countries [6]. Usually, the higher the socio-demographic index (SDI), the lower the disease burden [7].

There are numerous risk factors for musculoskeletal disorders, including ergonomic stressors, medical level factors, occupational risk factors [8], and individual health habits. Among these, occupational factors are the most significant influencing factor [9]. Repetitive high-intensity exercise, weightlifting, non-neutral posture, and vibration are all examples of ergonomic stressors [10]. Inadequate medical management [11] and low medical priority are two medical-level factors [12]. The occupational risks defined by the GBD mainly include six risk categories: carcinogens; asthma; occupational particulate matter, games, and fuels (PGFs); noise; injury; and ergonomic factors [13]. The GBD has included evidence assessment risk results that are consistent with the belief of the World Cancer Research Foundation (WCRF), and this evidence is biologically reasonable [14]. The occupational risks of musculoskeletal disorders are mainly manifested in some ergonomic factors related to work. Work stress, the frequency of work breaks [15], repetitive operations, and the work environment are the most common occupational risk factors [7]. Individual health habits factors primarily refer to low levels of physical activity [16], being sedentary, smoking, and standing for long periods. Of these, smoking is a particular factor affecting musculoskeletal disorders. Many published studies have examined the relationship between smoking and musculoskeletal disorders. In the context of musculoskeletal injuries, smoking is associated with higher rates of tissue hypoxia, slower wound healing, impaired blood flow, and postoperative healing complications [17]. Smoking reduces bone density and increases the risk of fractures or tendon injuries, and smoking increases the risk of perioperative complications, fracture nonunion and delayed union, infection, and soft tissue and wound healing complications [18]. There is also an association between smoking and the likelihood of developing long-term disability after musculoskeletal hospitalization [19]. The incidence of this disease significantly increased during the COVID-19 pandemic, mainly due to a lack of exercise [20]. Among all risk factors, tobacco, a high body mass index, kidney dysfunction, and occupational risks are the main factors leading to disability-adjusted life years (DALYs) of global musculoskeletal disorders [7]. In 2019, tobacco, a high body mass index, kidney dysfunction, and occupational risks accounted for 10.29 million DALYs, 7.44 million DALYs, 2300 DALYs, and 15.31 million DALYs, respectively, in the world [7]. These four risk factors explained about 22.16% DALYs of global musculoskeletal disorders [7].

In 2017, musculoskeletal disorders were ranked fifth worldwide for DALYs and first in years lost (YLD) due to disability globally [21]. Two of the top ten major factors, lower back pain and other musculoskeletal diseases, contributing to the increase in the global disease burden were musculoskeletal disorders [22]. Moreover, research has shown that the disease is prevalent at all ages from adolescence to old age [22], resulting in a high demand for rehabilitation services for musculoskeletal disorders [1]. Currently, there is a great deal of interest in research on musculoskeletal disorders from both government and corporate sources. As the country with the largest population in the world, China’s burden of musculoskeletal disorders has a large impact on the world average.

Like other countries in the world, there are six common types of musculoskeletal disorders in China: rheumatoid arthritis (RA), osteoarthritis (OA), lower back pain (LBP), neck pain (NP), gout, and other MSK diseases [7]. In 2017, the disability rate of musculoskeletal disorders in China ranked second [23]. Compared with the other 19 countries in the G20, China ranked second in the number of DALYs for musculoskeletal disorders [24]. It can be seen that the severity of musculoskeletal disorders in China has been increasing in recent years. Musculoskeletal disorders lead to physical impairment, psychological impairment and other problems, resulting in reduced productivity and increased medical costs that impose enormous health and economic burdens on individuals, communities and society [25]. According to reports, the prevalence of musculoskeletal disorders in various occupational groups in China is 20.00% to 90.00%, and it is as high as 90.00% in some industries [26]. For instance, the prevalence of NP among automobile assembly workers was found to reach 64.10% [27] and the prevalence rate of nursing population was found to be as high as 91.40% [28]. In this regard, the number of people suffering from musculoskeletal disorders in China is enormous. To the best of our knowledge, few economic burden studies have been conducted on musculoskeletal disorders in China, but it can be inferred from economic burden studies in other countries and regions of the world that the economic burden of musculoskeletal disorders in China is equally severe [29]. Disability and medical costs from musculoskeletal disorders in China could have major global health and economic impacts [25]. As life expectancy in China increases, so does the risk of chronic disease, and there is an urgent need for a large population-wide study of musculoskeletal disorder trends and key risk factors in China.

By combing through existing research, we found that most studies focused on musculoskeletal disorders in specific occupational groups [30] and there was a lack of analysis of the burden of disease in large populations and its attributable factors. In addition, few studies have focused on the burden of the disease in China. To fill this void, we aimed to analyze disability-adjusted life years (DALYs), death, incidence, and risk factors of musculoskeletal disorders in China from 1990 to 2019. We also predicted trends of incidence rates in China over the next 25 years for different age groups. The findings might inform relevant health strategies and contribute to the effective allocation of healthcare resources for disease prevention and control. This research not only provides clear messages on musculoskeletal disorder burden and its risk factors in China but also identifies key policy-maker targets for musculoskeletal disorders in China. In addition, our results might help in the formulation of relevant health policies and the effective allocation of healthcare resources for the prevention and control of musculoskeletal disorders.

## 2. Materials and Methods

### 2.1. Data Source

This study focused on analyzing the burden of musculoskeletal disorders and its risk factors in China and forecasting future incidence based on the incidence data and the serial interval. All the detailed data, including the Crude Incidence Rate (CIR), Crude Death Rate (CDR), Crude DALY Rate, Age-Standardized Incidence Rate (ASIR), Age-Standardized Death Rate (ASDR), Age-Standardized DALY Rate and population data associated with the disease, were obtained from the GBD 2019 database (https://vizhub.healthdata.org/gbd-results/, accessed on 31 August 2022), which was searched by sex and age from 1990 to 2019 [31]. The GBD 2019 database, developed by the Institute for Health Metrics and Evaluation (IHME) at the University of Washington, has disease statistics from more than 200 countries and regions around the world, and it is committed to providing normative and comparable measurements of important health issues on a global scale in order to improve the health system and reduce health disparities [7]. Each type of data in the GBD database is scientifically rigorous and identified through systematic reviews of government and international organization websites, as well as published studies and reports [32]. According to the GBD 2019 database, risk factors and predictions were only measured for those aged 30–84 years old. People under the age of 30 were excluded because the incidence of musculoskeletal disorders in this group of people is significantly lower than the average level (6422.37/100,000). Additionally, the influence of risk factors on musculoskeletal disorders in this group of people is almost zero. The reason for excluding people over 85 years old is that the incidence rate of this population is significantly opposite to the overall trend, showing a rapid downward trend. However, the general consensus of research is that the risk of musculoskeletal disorder increases with age, which is also verified in pathology [7]. Moreover, considering that the GBD data are based on a meta-regression of the literature and disease statistics released by governments using the DisMod model, there are statistical errors due to data accessibility and other issues. For these reasons, we excluded people over 85 years old. Finally, the trend forecast of disease incidence in China in the next 25 years is presented with the aim of providing future musculoskeletal disorder prevention focal points in order to improve health systems and reduce health disparities [22].

### 2.2. Statistical Analysis

Figure 1 shows the analysis roadmap of this study. We performed time-series analysis, the statistical analysis of risk factors, age–period–cohort analysis, and prediction analysis for musculoskeletal disorders. The time-series analysis was used to examine the Crude Incidence Rate (CIR), Crude Death Rate (CDR), Crude DALY Rate, Age-Standardized Incidence Rate (ASIR), Age-Standardized Death Rate (ASDR), and Age-Standardized DALY Rate from 1990 to 2019 in order to show the temporal development trend of musculoskeletal disorders in China. Risk factor analysis was used to study the effects of tobacco, a high body mass index, kidney dysfunction, and occupational risks on the ASDR and Age-Standardized DALY Rate. Age–period–cohort analysis was used to research the separate effects of age, period, and cohort on the incidence of musculoskeletal disorders from 1990 to 2019. Prediction analysis was used to examine the incidence trends of musculoskeletal disorders in China from 2020 to 2044. The specific analysis methods are as follows.

First, a time-series analysis of the burden of musculoskeletal disorders in China was carried out. Considering that the disease is strongly influenced by aging [33], crude and age-standardized data were counted separately. The Crude Incidence Rate, Crude Death Rate, Crude DALY Rate, Age-Standardized Incidence Rate, Age-Standardized Death Rate, and Age-Standardized DALY Rate were calculated for men, women, and overall. Furthermore, time trends and their significance were assessed using joinpoint regression methods, which are used to describe trends in data over an entire observation period and to calculate specific time points when trends changed [34]. To examine changes in the Age-Standardized Rate (ASR), the annual percentage change (APC) was calculated using a joinpoint regression model with natural log-transformation rates and selected linkage points. To determine the direction and extent of this overall disease trend, the average APC (AAPC) was also assessed from 1990 to 2019. The AAPC can be summarized using segmented APCs to compare rates of change over time and to determine long-term trends in rates of change, even if they are unstable [35]. In this study, the natural logarithm of the ASR was chosen as the response variable and the year of notification was used as the independent variable.

Second, we conducted a descriptive analysis of temporal and age trends in risk factors for musculoskeletal disorders in China. The GBD database estimates mortality, morbidity, prevalence, life years lost, disability life years, and DALYs for 87 risk factors in 204 countries and territories from around the world. These risk factors are in accordance with the World Cancer Research Fund criteria [36]. We conducted a statistical analysis of all risk factors that may contribute to musculoskeletal disorders in China through the GBD database. The results showed that tobacco, a high body mass index, kidney dysfunction and occupational risks were the attributable risk factors leading to the death and DALYs of musculoskeletal disorders in China. Therefore, we analyzed temporal trends in the contribution of these four risk factors from 1990 to 2019. For the Chinese data in the GBD database, the extent of disease risk was most representative for people aged 30 to 84 years, so we focused on trends for people aged 30 to 84 years. Additionally, attention was paid to the age differences between risk factors in 2019.

Next, we conducted age–period–cohort (APC) analyses for men and women, and we combed the raw data for the model requirements. Age–period–cohort (APC) analysis was used to estimate independent trends for age, period, and cohort effects from 1990 to 2019. Age effects represent different risks of outcomes associated with different age groups; period effects represent changes in outcomes over time, simultaneously affecting all age groups; and cohort effects are related to changes in outcomes for groups of individuals with the same birth year [37]. The first step was to divide people aged 30 to 84 years into 11 groups (30–34, 35–39, 40–44, 45–49, 50–54, 55–59, 60–64, 65–69, 70–74, 75–79 and 80–84), with each group separated by 5 years. Second, the same 5-year intervals were divided into six groups (1990–1994, 1995–1999, 2000–2004, 2005–2009, 2010–2014, and 2015–2019) throughout the observation period from 1990 to 2019. Finally, 16 birth cohorts were obtained by subtracting age (1907–1911, 1912–1916, 1917–1921……1972–1976, 1977–1981, and 1982–1986). We used the natural logarithm of disease incidence as the dependent variable and selected the median of these datasets as the independent variable to separately calculate age, period, and cohort effects. Because age, period and cohort have a perfectly linear relationship, it was not possible to identify the model. To overcome this multicollinearity problem, an intrinsic estimator (IE algorithm) was used in this study. This statistical approach has also been widely used in many published papers on the incidence of multiple diseases in global epidemics [9].

Finally, we conducted a prediction study of the incidence rate in different age strata. In order to more accurately predict the future development trend, we selected different time periods to conduct research. Predictions were made for the time groups of 30 years, 29 years, 28 years, 27 years, and up to 25 years, and it was found that the results of the 25 year group were significant and its data quality was the highest. Therefore, for the accuracy of the data, we conducted a forecast analysis for the next 25 years for the data from 1995 to 2019. The Bayesian APC model is well-suited to analyzing the prediction of age-stratified disease incidence rates. Therefore, we used the Bayesian APC model to predict trends in changes in the characteristics of the population aged 30 to 84 years from 2020 to 2044. The population was divided into 11 subgroups, 30–34, 35–39, 40–44, 45–49, 50–54, 55–59, 60–64, 65–69, 70–74, 75–79 and 80–84 years. Incidence rates were predicted for 11 age groups based on 25-year (1995–2019) time-series data on the incidence of the disease in China and 25-year time-series data on the Chinese population.

### 2.3. Software

The Joinpoint regression program (version 4.9.0.0) was used to analyze the trends of incidence, death, and DALY rates of the disease from 1990 to 2019. *p*-values of less than 0.05 were considered statistically significant. The GBD 2019 database identified risk factors for musculoskeletal disorders, and we performed descriptive analyses by sex and age. For APC model analyses and graphs, APC fit in Stata (version 13.0) was used. The Bayesian APC was modeled using the BAMP package in R (version 4.1.12) to predict the incidence in the next 30 years (2020–2049). All figures were drawn with OriginPro (version 2020b).

## 3. Results

### 3.1. Musculoskeletal Disorder Burden in China

Figure 2 shows the trends in the incidence rate, death rate and DALY rate for musculoskeletal disorders by gender in China from 1990 to 2019. As can be seen, there were significant differences between the crude data and the data standardized by age. Overall, the CIR, CDR and Crude DALY Rate for the disease showed increasing trends from 1990 to 2019. In particular, the CIR and Crude DALY Rate for the disease showed significant downward trends from 1990 to 1994 and consistent upward trends from 1994 to 2019. There was an apparent turnaround around 1994. Specifically, the CIR and Crude DALY Rate for the disease decreased by 11.40% and 5.53%, respectively, from 1990 to 1994. The CIR, CDR, and Crude DALY Rate for the disease significantly increased by 34.81%, 98.95%, and 50.08%, respectively, from 1994 to 2019. Interestingly, after age standardization, the trends clearly diverged. From 1990 to 2019, the ASIR and Age-Standardized DALY Rate for the disease trended slightly downwards and the ASDR trended slightly upwards. Specifically, the ASIR decreased from 4580.28/100,000 in 1990 to 3764.99/100,000 in 2019, the ASDR increased from 1.03/100,000 in 1990 to 1.20/100,000 in 2019, and the Age-Standardized DALY Rate decreased from 1687.59/100,000 in 1990 to 1585.44/100,000 in 2019. All detailed data are shown in Appendix A.

In terms of gender, the overall trend in the burden of disease for both men and women was in line with the trend for the total population. Particularly, the incidence rate, death rate, and DALY rate were found to be higher for women than men. In terms of crude data, the growth rates of the CIR and Crude DALY Rate were similarly higher for women than men. In particular, the growth rate of the CDR was slightly higher for men than women. Specifically, between 1994 and 2019, the CIR, CDR, and Crude DALY Rate for this disease among men increased by 28.83%, 106.17%, and 45.30%, respectively, while the CIR, CDR, and Crude DALY Rate increased by 39.30%, 93.86%, and 52.91%, respectively, among women. After age standardization, the fluctuating trends in the ASDR, ASIR and Age-Standardized DALY Rate for musculoskeletal disorders flattened down for both men and women from 1990 to 2019. From 1990 to 2019, the ASIR for both men and women showed a downward trend, decreasing by 17.73% and 18.29%, respectively. Regarding the ASDR, both men and women showed an upward trend, rising by 5.49% and 22.88%, respectively. In terms of the Age-Standardized DALY Rate, both men and women presented a downward trend, decreasing by 5.26% and 7.12%, respectively.

### 3.2. Joinpoint Regression Analysis of the Disease Burden of Musculoskeletal Disorders in China

Table 1 and Table 2 show the results of the joinpoint regression analysis for the crude data and age-standardized data of musculoskeletal disorders from 1990 to 2019. As can be seen, there were significant changes in the trend after standardized by age. The CIR generally showed a decreasing trend followed by an increasing trend (AAPC = 0.6). It rapidly declined from 1990 to 1994 (AAPC = −3.1) and steadily increased from 1995 to 2019. Among men, similar to the results for the whole population, the CIR trended down and then up throughout the entire research period (AAPC = 0.5), with 1994 being the turning point. Likewise, the CIR for women maintained the same trend (AAPC = 0.7), with a slightly more significant increase than the overall and men trends. Concerning the CDR, like the CIR, it generally showed a trend of first falling and then rising (AAPC = 2.5), initially decreasing from 1990 to 1993, then beginning to significantly rise from 1993 to 1998, followed by a sharp climb from 1998 to 2001 (AAPC = 7.9) and an upward trend thereafter. Among men, similar to the overall trend, it started to decline until 1994, followed by a sustained increase from 1994 to 2019 (AAPC = 2.1). Similarly, the trend for women also declined and then increased (AAPC = 2.6). Finally, for the Crude DALY Rate, the trends for men, women and the whole population were first downward and then upward.

Interestingly, the trend of the disease burden significantly changed after age standardization, with the ASIR, ASDR and Age-Standardized DALY Rate all levelling off. Both the ASIR and Age-Standardized DALY Rate showed slight downward trends. Specifically, the overall downward trend in the ASIR gradually weakened, with an AAPC of −0.7 for both men and women. The ASDR generally showed a fluctuating upward trend (AAPC = 0.5). It went down at first until 1993, when it began to dramatically increase until 2005, followed by a slight downward trend from 2005 to 2015 and then a slowly rising trend from 2015 to 2019. Similarly, the ASDR demonstrated fluctuating patterns for men (AAPC = 0.2) and women (AAPC = 0.7), with a slightly greater increase than for men. Finally, the Age-Standardized DALY Rate showed an overall decreasing trend. The AAPC for both men and women was −0.2.

### 3.3. Difference in Attributable Risk Factors

Through the GBD statistical analysis of all risk factors, we concluded that tobacco, a high body mass index, kidney dysfunction and occupational risks were the leading four risk factors contributing to musculoskeletal disorder deaths and DALYs in China. From 1990 to 2019, the average annual DALYs of tobacco, a high body mass index, kidney dysfunction, and occupational risks were 108.75/100,000, 38.05/100,000, 1.87/100,000, and 203.80/100,000, respectively. According to the GBD database, during the 30 years from 1990 to 2019, death from musculoskeletal disorders in China was overwhelmingly attributable to tobacco. The trend first showed a decline, then a minimum in 1997 (ASDR both = 0.061/100,000), followed by a rise peaking around 2005 (ASDR both = 0.093/100,000), and then a steady decline. The four risk factors attributed to DALYs were tobacco, a high body mass index, kidney dysfunction and occupational risks. Of the four risk factors, occupational risks had the highest impact, followed by tobacco, a high body mass index, and kidney dysfunction. The effects of kidney dysfunction and a high body mass index showed increasing trends year by year. In contrast, the impact of tobacco and occupational risks showed a decreasing trend. Regarding gender, the risk factors for musculoskeletal disorders in women were ranked from high to low as follows: occupational risks, a high body mass index, tobacco and kidney dysfunction. Among men, the risk factors for musculoskeletal disorders in descending order were occupational risks, tobacco and a high body mass index. It can be seen that the effects of tobacco and kidney dysfunction on musculoskeletal disorders were higher in men than in women; in particular, the effects of smoking in men were much higher than those in women. In contrast, the effects of a high body mass index and occupational risks were higher in women than in men (Figure 3). The data are shown in Appendix A.

In terms of age, according to the GBD data, the impact of musculoskeletal disorders in China was concentrated in the 30 to 84 year age group. Similarly, tobacco was found to be a determinant of mortality among those aged 30–84 years. In terms of the degree of impact leading to death, the age distribution was shown to be essentially the same for both men and women, with a trend of increasing impact with age. Differences in gender distribution began to appear around the age of 50. We found that as age increases, the effect of tobacco on the ASDR was much greater in men than in women. In terms of the effect on disability-adjusted life years, the effects of tobacco and kidney dysfunction showed positive correlations with age, while the effects of a high body mass index and occupational risks tended to increase and then decrease with age. The effects of a high body mass index and occupational risks tended to increase and then decrease with age. Specifically, the effect of occupational risks on disability-adjusted life years peaked at 50–54 years of age and the effect of a high body mass index on disability-adjusted life years peaked at 65–69 years of age. Overall, among those aged 30–65 years, occupational risks had the most significant effect on disability-adjusted life years and kidney dysfunction had the slightest effect. Among those aged 65 to 84, tobacco was the most significant factor affecting disability-adjusted life years (Figure 4). The data are shown in Appendix A.

### 3.4. Age–Period–Cohort (APC) Model Analysis of Musculoskeletal Disorders Incidence in China

As shown in Figure 5, the effect of age on the incidence of musculoskeletal disorders showed an increasing trend in China. The period effect generally tended to flatten. On the other hand, the cohort effect tended to decrease throughout the timeline. The age effect coefficient for both men and women rapidly increased and then gradually slowed down. For those aged 30–45 and 70–84, the age effect coefficients were larger for men than for women, while the exact opposite was true for those aged 50 to 69. Interestingly, for those aged 70 and over, the effect coefficient for men significantly rose, and the trend slowed down for women. The period effect coefficients for both men and women gradually leveled off. In addition, the cohort effect coefficients for both sexes trended downwards, with a greater rate of decline for women, who had larger cohort effect coefficients than men before 1966. However, after 1966, women had smaller cohort effect factors than men and the rate of decline accelerated for women (Figure 5). The data are shown in Appendix A.

### 3.5. Prediction Based on the Bayesian Age Cohort Model

The results showed a significant change in the trend of musculoskeletal disorders in China from 2020 to 2044 depending on the age group. Those aged 35–59 showed a decreasing trend, while those aged 30–34 and those aged 65–84 showed slightly increasing trends. The growth rates were 1.22%, 2.70%, 4.66%, 4.02%, and 1.99% for the 30–34, 65–69, 70–74, 75–79, and 80–84 year age groups, respectively. The rates of decrease were 0.77%, 3.08%, 4.02%, 3.63%, and 2.60% for the 35–39, 40–44, 45–49, 50–54, and 55–59 year age groups, respectively (Figure 6). It is noteworthy that the highest rate of increase was recorded in the 70–74 year age group at 4.66% and the lowest rate of increase was recorded in the 30–34 year age group at 1.22%. Conversely, the highest rate of decline was seen in the 45–49 year age group at 4.02% and the lowest rate of decline was seen in the 35–39 year age group at 0.77%. Overall, as age increased, so did the incidence rate. The data are shown in Appendix A.

## 4. Discussion

To our knowledge, this is the first comprehensive study to present the latest analysis of the musculoskeletal disorder burden, risk factors, and future predictions of disease trends in different age groups in China. Our study found that the CIR, CDR and Crude DALY Rate for this disease have substantially increased over time since the beginning of the 21st century. This is in line with the global trend of this disease, and the current high rehabilitation demand for musculoskeletal disorders has become a global public health issue [38]. From the study of risk factors for this disease, we concluded that this trend might be partly attributed to increased occupational stress, poor health habits, and high body weight due to modern diets. Especially in low- and middle-income countries, the burden from musculoskeletal disorders has substantially increased in recent years due to limited health systems and national economic capacity [11]. According to the data of GBD2019, Barbados, Bahamas and Honduras had the highest ADRs. In terms of change from 1990 to 2019, Zambia had the largest increases in the ASIR and Age-Standardized DALY Rate. 

As musculoskeletal disorders are age-dependent diseases, their disease burden is closely related to the demographic structure and the degree of population aging [39]. Therefore, once the effects of population growth and aging are removed through age-standardized rates, the overall health status produces different outcomes. Our study found that after age standardization, the development trend of disease burden was significantly different, changing to a flat progression. In addition to the impact of population aging [24], this may also be related to the age distribution of musculoskeletal disorders in China. According to the APC study, middle-aged and elderly groups are at high risk, which may be related to their lower awareness of personal health and the natural effects of age on bones [40]. Awareness of the importance of musculoskeletal health is increasing among younger populations. At the same time, with gradual improvements in sanitation conditions, the younger generation is enjoying better sanitation resources. In addition, it is expected that the age structure of China’s population will change from 2023 to 2050, and an accelerated aging model will begin, with the proportion of the population aged 80 and above increasing year by year [41]. It has been argued that with an increasingly aging population, the burden of this disease will face significant challenges in the future and the need for rehabilitation will substantially increase [38].

In terms of gender differences, we found that the burden of musculoskeletal disorders was more severe in Chinese women than men, which is the same as the general development trend of the disease in the world [7]. In 2019, the number of deaths of women due to this disease worldwide was 2.24 times more than that of men, and the DALYs were 1.45 times higher than those of men [7]. This phenomenon may be related to female physiology, as males are naturally much more muscular and more musculoskeletal protective than females [42]. Moreover, pregnancy and childbirth increase the burden of musculoskeletal disorders in women [43]. Our study found that the incidence, death and DALYs of musculoskeletal disorders are significantly higher in women than in men. Therefore, it is vital to pay more attention to musculoskeletal disorders in women and to enhance health awareness. In addition, we found significant gender differences in the risk factors for musculoskeletal disorders in China. The effects of tobacco and kidney dysfunction on musculoskeletal disorders were found to be higher in men than in women; in particular, the effects of smoking were much higher in men than in women. The 2018 China Health Literacy Survey (2018 CHLS) reported a significantly higher prevalence of smoking among Chinese men than women [44]. However, a high body mass index and occupational risks were shown to have a higher effect on women than men. In addition, our APC study found that men were at greater risk for this disease than women for those aged 30–45 and 70–84 years, while women were at greater risk for those aged 50–69 years. Cohort effect coefficients showed that risk factors became less influential and health improved more in women than in men as the years progressed. It is necessary to focus on preventing male musculoskeletal disorders and improving male health awareness.

From the perspective of risk factors based on GBD data, this study found that during the 30 years from 1990 to 2019, the most significant risk factor of musculoskeletal disorders in China was tobacco and the most significant risk factors of musculoskeletal disorders in China for disability-adjusted life years were tobacco, a high body mass index, kidney dysfunction and occupational risks. Moreover, the effects of kidney dysfunction and a high body mass index have been increasing in recent years, while the effects of tobacco and occupational risks have been decreasing. This indicates that the trend of musculoskeletal disorders due to internal causes is gradually increasing and the influence of external factors is gradually decreasing. In the long run, early screening and health intervention policies can reduce the disease burden. From the age perspective, we found that people aged 50–54 were more likely to have disability-adjusted life years for musculoskeletal disorders due to occupational risks. We found also that people aged 65–69 need to pay more attention to the impact of a high body mass index. Exposure to a high BMI has been increasing nationwide in recent years, and weight management can effectively reduce the impact of musculoskeletal disorders [24].

The predicted results showed that the incidence rate increases with age. Our study predicts that the incidence will show different trends with age from 2020 to 2044. The populations aged 30–34 and 60–84 showed a clear upward trend, while those aged 35–59 showed a downward trend. Compared with other age groups, the incidence will rapidly increase in 75–94 years old people. Similar to the global response to the disease, reducing occupational risk, limiting tobacco use, and lowering one’s body mass index can help reduce the burden of this disease in China [7]. As China enters an aging society, tertiary preventive measures must be incorporated into early health interventions for this disease.

There were some limitations of our study. Firstly, we could not conduct a more specific regional comparative analysis of musculoskeletal disorders across regions in China due to a lack of provincial data. Previous studies have only analyzed trends in local disease burden for a single occupation and region. In Guangdong Province, work-related musculoskeletal disorders (WMSDs) among sonographers are closely related to work stress, and reducing workload was considered key to prevention and control [45]. In Xinjiang Province, musculoskeletal disorders among nurses were strongly associated with shift and work breaks [46]. There has been no regional analysis for the entire population. In China, local customs widely vary. Additional micro-regional analysis is needed to guide disease prevention and control policy. Secondly, our research comprised a secondary data analysis of the 2019 GBD data. Therefore, the accuracy of the results heavily depends on the quality and quantity of the data available. Future research should focus on comparative regional analyses between different provinces in China, which could contribute to the mastering of regional variations to guide appropriate health policies and programs, as well as facilitate the realization of Health China 2030.

## 5. Conclusions

The burden of musculoskeletal disorders in China is on the rise. Compared with younger people, the disease burden is relatively high in older people, mainly due to tobacco, a high body mass index, kidney dysfunction and occupational risks. In addition, the prevalence is expected to further increase in the working population aged 30–34 years and older people aged 60–84 years in 2020–2044, and attention should be focused on these age groups. The incidence is most rapidly increasing in older people aged 75–94 years and should be of particular concern. Women have a higher burden of disease than men, mainly due to weight reasons and work exposure. Tobacco is the most significant risk factor for musculoskeletal disorders in men. Disease control policies and early screening should focus on women and the elderly, with various health interventions based on risk factors (e.g., health awareness, early screening, timely access to health care, tobacco use, weight management, and environmental protection at work) to achieve the effective prevention of musculoskeletal disorders at low cost. As a typical example of a rapidly developing country, China’s recent trends in the burden of musculoskeletal disorder can serve as an important basis for the development of health policies related to this disease in developing countries worldwide.

## Figures and Tables

**Figure 1 ijerph-20-00840-f001:**
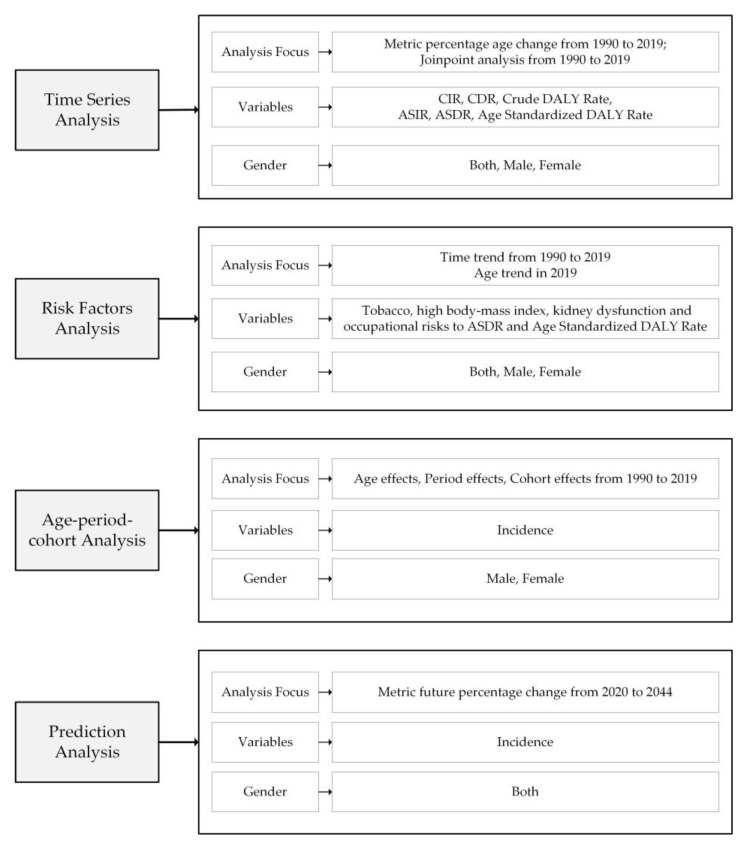
Analysis roadmap of this study.

**Figure 2 ijerph-20-00840-f002:**
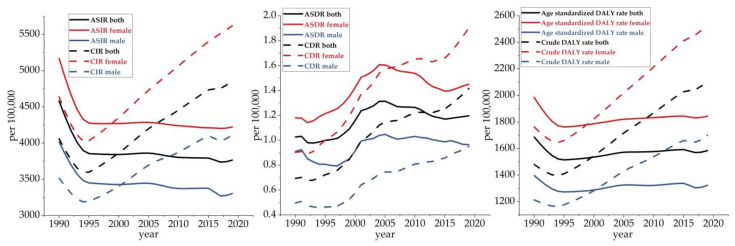
Trends of musculoskeletal disorder burden in China from 1990 to 2019.

**Figure 3 ijerph-20-00840-f003:**
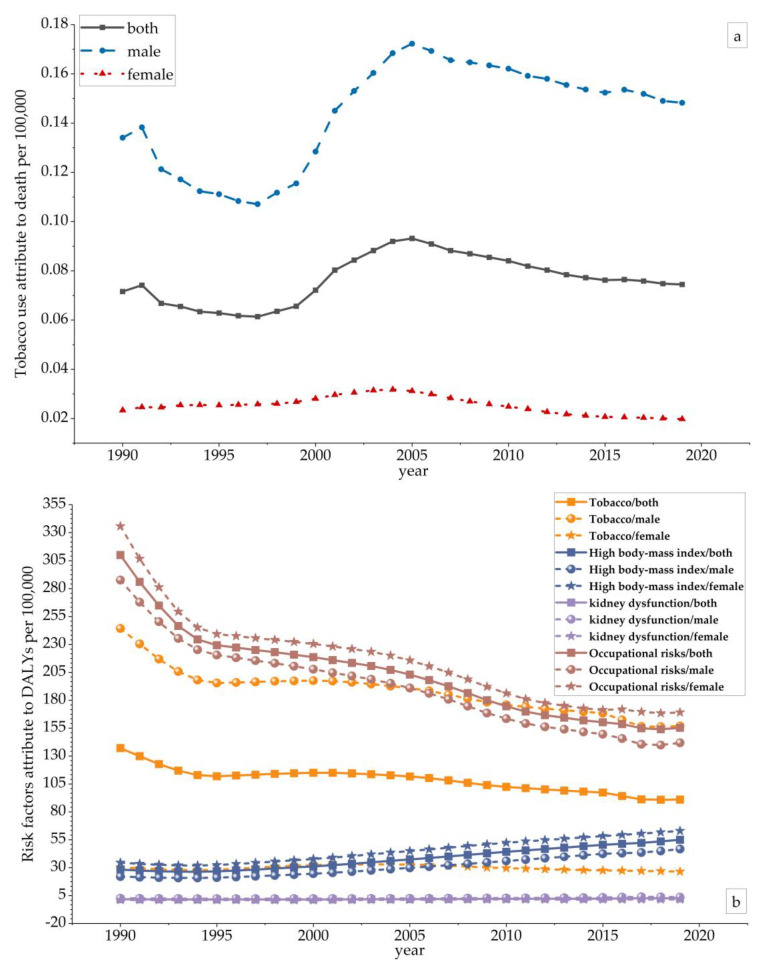
The variation trends of the Age–Standardized Death Rate (ASDR) and Age–Standardized DALY Rate of risk factors in different genders over 30 years. (**a**) The variation trends of the ASDR in different genders over 30 years; (**b**) the variation trends of the Age–Standardized DALY Rate of risk factors in different genders over 30 years.

**Figure 4 ijerph-20-00840-f004:**
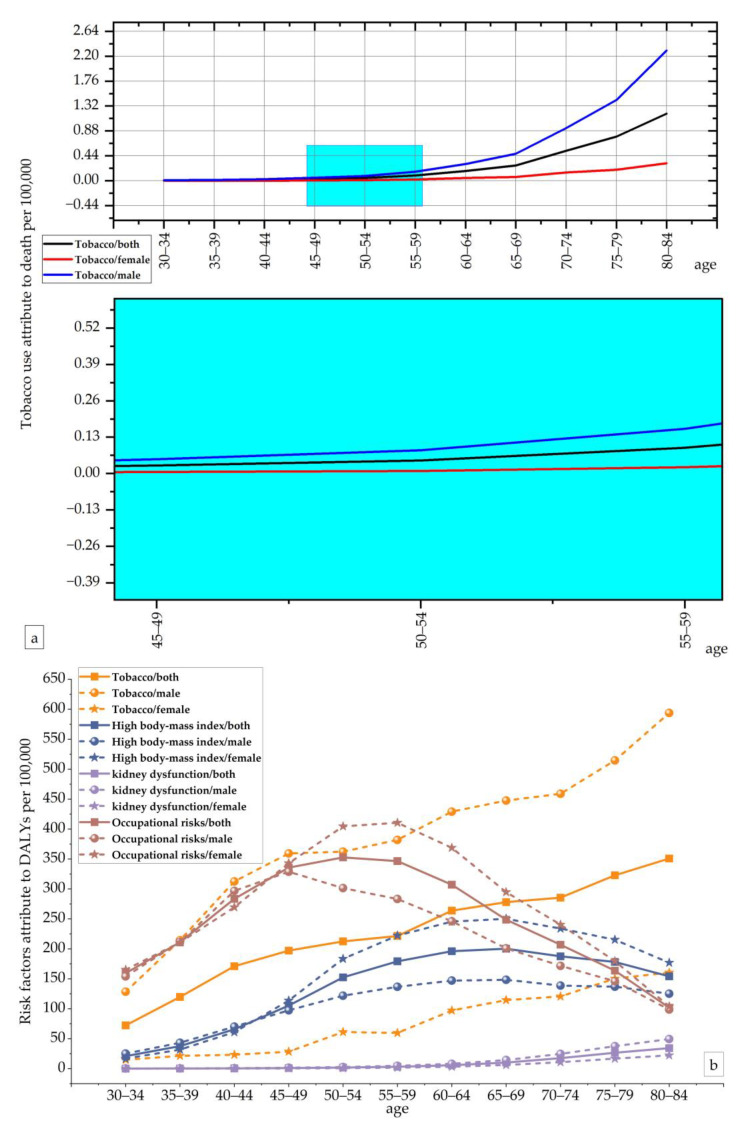
The variation trend of the ASDR and Age–Standardized DALY Rate of risk factors in different genders and age groups in 2019. (**a**) The variation trends of the ASDR in different genders and age groups in 2019; (**b**) the variation trends of the Age–Standardized DALY Rate of risk factors in different genders and age groups in 2019.

**Figure 5 ijerph-20-00840-f005:**
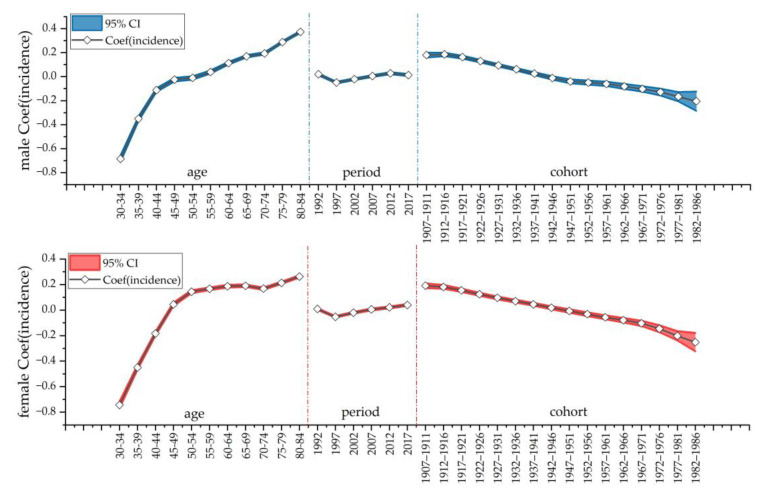
Age–period–cohort (APC) model analysis of musculoskeletal disorder incidence among females and males in China.

**Figure 6 ijerph-20-00840-f006:**
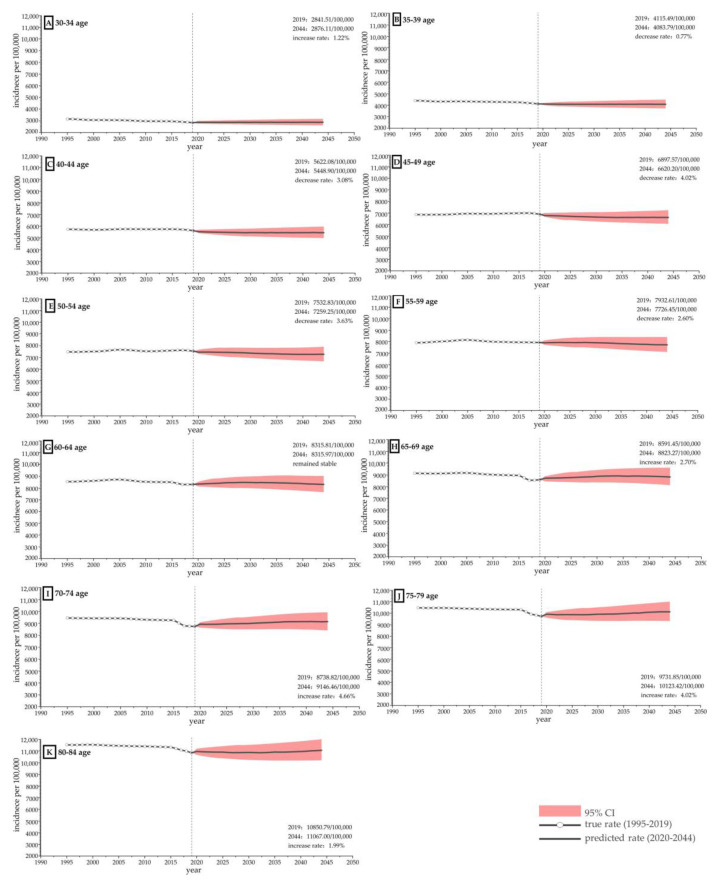
Prediction of musculoskeletal disorder incidence among females and males in China from 2020 to 2044 based on the Bayesian APC model. (**A**) Prediction of incidence rate of musculoskeletal disorder in people aged 30–34 in China; (**B**) Prediction of incidence rate of musculoskeletal disorder in people aged 35–39 in China; (**C**) Prediction of incidence rate of musculoskeletal disorder in people aged 40–44 in China; (**D**) Prediction of incidence rate of musculoskeletal disorder in people aged 45–49 in China; (**E**) Prediction of incidence rate of musculoskeletal disorder in people aged 50–54 in China; (**F**) Prediction of incidence rate of musculoskeletal disorder in people aged 55–59 in China; (**G**) Prediction of incidence rate of musculoskeletal disorder in people aged 60–64 in China; (**H**) Prediction of incidence rate of musculoskeletal disorder in people aged 65–69 in China; (**I**) Prediction of incidence rate of musculoskeletal disorder in people aged 70–74 in China; (**J**) Prediction of incidence rate of musculoskeletal disorder in people aged 75–79 in China; (**K**) Prediction of incidence rate of musculoskeletal disorder in people aged 80–84 in China.

**Table 1 ijerph-20-00840-t001:** Log-transformed joinpoint trends of musculoskeletal disorder crude rates by sex in China.

Measure	Sex	Trend 1		Trend 2		Trend 3		Trend 4		Trend 5		Trend 6		1900–2019 AAPC (95% CI)
		Years	APC	Years	APC	Years	APC	Years	APC	Years	APC	Years	APC	
Crude Incidence Rate	Both	1990–1994	−3.1 *	1994–1998	1.2 *	1998–2005	1.7 *	2005–2014	1.2 *	2014–2019	0.7 *	NA	NA	0.6 * (0.6~0.7)
	Female	1990–1994	−3.7 *	1994–2009	1.6 *	2009–2019	1.2 *	NA	NA	NA	NA	NA	NA	0.7 * (0.7~0.7)
	Male	1990–1994	−2.5 *	1994–1998	1.1 *	1998–2005	1.6 *	2005–2014	1.0 *	2014–2019	0.2	NA	NA	0.5 * (0.4~−0.7)
Crude Death Rate	Both	1990–1993	−1.0	1993–1998	3.1 *	1998–2001	7.9 *	2001–2005	3.8 *	2005–2015	0.8 *	2015–2019	3.0 *	2.5 * (2.0~2.9)
	Female	1990–1992	−1.1	1992–1998	3.9 *	1998–2001	6.9 *	2001–2005	3.8 *	2005–2015	0.5 *	2015–2019	3.3 *	2.6 * (2.2~3.0)
	Male	1990–1994	−2.4 *	1994–1997	0.7	1997–2003	7.2 *	2003–2019	1.7 *	NA	NA	NA	NA	2.1 * (1.6~2.7)
Crude DALY Rate	Both	1990–1993	−1.9 *	1993–1996	0.7	1996–2005	2.1 *	2005–2014	1.7 *	2014–2019	0.9 *	NA	NA	1.2 * (1.1~1.3)
	Female	1990–1993	−2.3 *	1993–1996	0.8 *	1996–2006	2.1 *	2006–2014	1.8 *	2014–2019	1.2 *	NA	NA	1.2 * (1.2~1.3)
	Male	1990–1994	−1.1 *	1994–1999	1.6 *	1999–2004	2.3 *	2004–2014	1.6 *	2014–2017	0.2	2017–2019	1.3 *	1.2 * (1.1~1.2)

Notes: AAPC, average annual percent change; APC, annual percent change; CI, confidence interval; NA, not applicable. * Significantly different from zero, *p*-value < 0.05.

**Table 2 ijerph-20-00840-t002:** Log-transformed joinpoint trends of musculoskeletal disorder age-standardized rates by sex in China.

Measure	Sex	Trend 1		Trend 2		Trend 3		Trend 4		Trend 5		Trend 6		1900–2019 AAPC (95% CI)
		Years	APC	Years	APC	Years	APC	Years	APC	Years	APC	Years	APC	
Age-Standardized Incidence Rate	Both	1990–1994	−4.1 *	1994–2005	−0.0	2005–2019	−0.2 *	NA	NA	NA	NA	NA	NA	−0.7 * (−0.7~−0.6)
	Female	1990–1994	−4.5 *	1994–2005	−0.0	2005–2019	−0.1 *	NA	NA	NA	NA	NA	NA	−0.7 * (−0.7~−0.7)
	Male	1990–1994	−3.7 *	1994–2006	−0.0	2006–2010	−0.5 *	2010–2014	0.2	2014–2017	−1.1 *	2017–2019	0.3	−0.7 * (−0.8~−0.6)
Age-Standardized Death Rate	Both	1990–1993	−2.0 *	1993–1998	1.2 *	1998–2001	6.0 *	2001–2005	1.7 *	2005–2015	−1.2 *	2015–2019	0.5	0.5 * (0.1~0.9)
	Female	1990–1992	−1.9	1992–1998	2.0 *	1998–2001	5.3 *	2001–2005	1.9 *	2005–2015	−1.5 *	2015–2019	0.8	0.7 * (0.3~1.1)
	Male	1990–1994	−3.5 *	1994–1998	0.1	1998–2001	6.9 *	2001–2004	1.9	2004–2019	−0.4 *	NA	NA	0.2 (−0.5~0.8)
Age-Standardized DALY Rate	Both	1990–1993	−3.0 *	1993–1996	−0.6 *	1996–2004	0.4 *	2004–2014	0.2 *	2014–2017	−0.4 *	2017–2019	0.4	−0.2 * (−0.3~−0.2)
	Female	1990–1994	−2.9 *	1994–2008	0.3 *	2008–2019	0.0	NA	NA	NA	NA	NA	NA	−0.2 * (−0.3~−0.2)
	Male	1990–1994	−2.2 *	1994–1999	0.2	1999–2004	0.6 *	2004–2014	0.1 *	2014–2017	−0.7	2017–2019	0.4	−0.2 * (−0.3~−0.1)

Notes: AAPC, average annual percent change; APC, annual percent change; CI, confidence interval; NA, not applicable. * Significantly different from zero, *p*-value < 0.05.

## Data Availability

The data used in this study are openly available in GBD 2019 at https://vizhub.healthdata.org/gbd-results/, accessed on 31 August 2022, reference number [7].

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
