# Peer review of "Musculoskeletal Disorder Burden and Its Attributable Risk Factors in China: Estimates and Predicts from 1990 to 2044"

_ijerph, 2023, doi:10.3390/ijerph20010840_

Round 1
Reviewer 1 Report
Dear author, congrats on your work.
Your work has enough merit to be published, however, there are some questions that should be solved before publication.
This paper has a lot of info, so it becomes hard for readers to follow the document. In the title, you abord musculoskeletal disorders, however, in a document you research a lot of variables furthermore. At this point is essential to put the whole document in the same line to turn the paper more readable.
Abstract:
Turn clearer the data about project incidence...is confusing. What is represented?
Introduction:
The authors should identify the type/frequency of musculoskeletal disorders in China and talk about the real impact of it on the economy.
Data Source:
“All the detailed data were obtained from the GBD 2019 database” – All data is from Global data or only from China? need to be more specific.
“According to the GBD 2019 database, risk factors and predictions were measured only for those aged 30-84 years old” – Why this age range? This age range option should be prior justified.
Reviewer 2 Report
The article This study aimed to assess the burden of musculoskeletal disorders in China from 1990 to 2019, risk factors and trends with simulation to 2044.
The paper is interesting and well written. To make it more comprehensive I would suggest Authors to
-elaborate more about possible mechanism how smoking may be ralated with musculoskeletal disorders. While the relationships between body weight, work-related risk, or even kidney disease seem plusible, smoking appears to be a marker of some other condition that has not been assessed?
-directly describe the strengths of this project
-describe or refer to the source how musculoskeletal disorders are defined by GBD consortium
